# New Copper Complexes with Antibacterial and Cytotoxic Activity

**DOI:** 10.3390/ijms241813819

**Published:** 2023-09-07

**Authors:** Adriana Corina Hangan, Roxana Liana Lucaciu, Alexandru Turza, Lucia Dican, Bogdan Sevastre, Emöke Páll, Luminița Simona Oprean, Gheorghe Borodi

**Affiliations:** 1Department of Inorganic Chemistry, Faculty of Pharmacy, “Iuliu-Hațieganu” University of Medicine and Pharmacy, 400012 Cluj-Napoca, Romania; acomsa6@yahoo.com (A.C.H.); loprean@umfcluj.ro (L.S.O.); 2Department of Pharmaceutical Biochemistry and Clinical Laboratory, Faculty of Pharmacy, “Iuliu-Hațieganu” University of Medicine and Pharmacy, 400012 Cluj-Napoca, Romania; roxanaluc@yahoo.com; 3National Institute for R&D of Isotopic and Molecular Technologies, 400293 Cluj-Napoca, Romania; turzaalex@yahoo.com (A.T.); borodi@itim-cj.ro (G.B.); 4Department of Medical Biochemistry, Faculty of Medicine, “Iuliu-Hațieganu” University of Medicine and Pharmacy, 400012 Cluj-Napoca, Romania; 5Paraclinic/Clinic Department, Faculty of Veterinary Madicine, University of Agricultural Science and Veterinary Medicine, 400372 Cluj-Napoca, Romania; bogdan.sevastre@usamvcluj.ro (B.S.); emoke.pall@usamvcluj.ro (E.P.)

**Keywords:** Cu^+2^ complexes, crystal structure, antibacterial, cytotoxicity

## Abstract

The discovery of a new non-toxic metal complex with biological activity represents a very active area of research. Two Cu^+2^ complexes, [Cu_4_(L1)_4_(OH)_4_(DMF)_2_(H_2_O)] (C1) (HL1 = N-(5-ethyl-[1,3,4]–thiadiazole–2-yl)-benzenesulfonamide) and [Cu(L2)_2_(phen)(H_2_O)] (C2) (HL2 = N-(5-(4-methylphenyl)-[1,3,4]–thiadiazole–2-yl)-naphtalenesulfonamide), with two new ligands were synthesized. The X-ray crystal structures of the complexes were determined. In both complexes, Cu^+2^ is five-coordinated, forming a CuN_2_O_3_ and CuN_4_O chromophore, respectively. The ligands act as monodentate, coordinating the metal ion through a single N_thiadiazole_ atom; for the two complexes, the molecules from the reaction medium (phenantroline, dimethylformamide and water) are also involved in the coordination of Cu^+2^. The complexes have a distorted square pyramidal square-planar geometry. The compounds were characterized by FT-IR and UV-Vis spectroscopy. Using the microdilution method, the antibacterial activity of the complexes was determined against four Gram-positive and two Gram-negative bacteria, with Gentamicin as the positive control. Cytotoxicity studies were carried out on two tumor cell lines (HeLa, DLD-1) and on a normal cell line (HFL1) using the MTT method and Cisplatin as a positive control. Flow cytometric assessment of apoptosis induced by the complexes on the three cell lines was also performed. Both complexes present in vitro biological activities but complex C2 is more active.

## 1. Introduction

One of the biggest public health problems of the 21st century is cancer, which, along with heart diseases and infectious diseases, remains among the top three causes of death worldwide. Each year, the American Cancer Society estimates the numbers of new cancer cases and deaths, and in 2023, 1,958,310 new cancer cases and 609,820 cancer deaths are projected to occur in the United States [1]. The incidence of cancer worldwide has increased annually to 19.3 million new cases and 10 million deaths. For the year 2040, more than 28.4 million cases of cancer are estimated [2]. Cancer—a term covering over 200 diseases—is today the second leading cause of death in Europe, with 1.3 million deaths and 3.5 million new cases per year. Without further action to reverse current trends, it is estimated that by 2035, cancer cases in Europe could have doubled in number, making the disease the leading cause of death in middle age, before cardiovascular diseases [3]. Despite the numerous discoveries of the last decades and therapies emerging innovations against cancer (molecular analysis, immunotherapy and personalized treatment), the fight against this disease continues [4,5,6,7].

The therapeutic potential of metal complexes in cancer therapy is of particular interest, as metals exhibit the following characteristics, such as redox activity, different coordination modes and reactivity towards organic substrate, essential in the design of antitumor providers [8,9]. They must be able to selectively interact with the biomolecular target and subsequently alter the cellular mechanism of proliferation [10,11].

Copper is an essential element for living organisms, being a cofactor for numerous enzymes, intervening in iron metabolism, hematopoiesis, porphyrin synthesis and in numerous metabolic processes. It is also an essential cofactor for a large number of proteins involved in redox reactions [12].

Unlike normal cells, tumor cells show a reduced vascularity that results in a low level of oxygen, a fact that explains invasion, metastasis and a metabolic shift towards an anaerobic process known as the Warburg effect [13]. As a result, tumor hypoxia can be exploited to develop new prodrugs that become active in the reducing environment of cancer cells. In this sense, copper as a metal becomes very attractive because it can exist in two different oxidation states in cells. The presence of hypoxia in cancer cells promotes the reduction of Cu^+2^ to Cu^+^, which is not possible in normal cells, and thus provides a therapeutic opportunity for copper compounds to target tumors [14]. The Cu^+^ ion, once formed, can catalyze the formation of reactive oxygen species (ROS) and reactive nitrogen species (RNS) to induce a state of pro-apoptotic oxidative stress [11].

For the success of antitumor strategies based on the administration of a copper complex, the set of donor atoms of the ligand is of crucial importance. It can modulate the chemical properties of the metal ion, the lipophilic/hydrophilic balance of the resulting complexes, the solubility in extracellular fluids as well as their ability to penetrate through the lipid bilayer membrane.

The antibacterial activity is probably the most well-known property and application of N-substituted heterocyclic sulfonamides, but some have a hypoglycemic, antithyroid, diuretic activity or inhibit carbonic anhydrase. Recent studies indicate new directions for the research of sulfonamides in order to broaden the antibacterial spectrum and to increase their efficiency [15,16].

From the point of view of coordinative chemistry, these molecules can function as a ligand due to the presence of N, O or S atoms, but also of other atoms belonging to the heterocyclic substituents. Depending on the desired therapeutic goal, the most diverse structures of sulfonamides can be designed. For example, for antitumor action, the inclusion of planar aromatic rings to facilitate the intercalation of metal complexes with sulfonamides between the base pairs of the DNA (deoxyribonucleic acid) molecule.

Microbial drug resistance has also become a serious medical problem, causing morbidity and mortality, which has attracted the attention of different researchers working on the discovery of antimicrobial drugs [17,18].

Sulfonamide ligands coordinated with Cu^+2^ can interfere with the biosynthesis of tetrahydrofolic acid, which is essential for bacterial metabolism [19,20]. Many sulfonamide Cu^+2^ complexes show antimicrobial activity against both types, Gram (+) (*Staphylococcus aureus*, *Bacillus subtilis*) and Gram (−) (*Escherichia coli*, *Pseudomonas aeruginosa*) bacteria [18,21].

In recent decades, numerous studies have demonstrated the antitumor and/or antibacterial action of Cu^+2^ complexes with N-substituted sulfonamides with remarkable results both in vivo and in vitro. The compounds obtained have a good bioavailability, are soluble in biological environments and have a relatively low toxicity [19,22,23,24].

The attention of our research group has been directed during the last 15 years to the synthesis, characterization and evaluation of the nuclease activity of some binary or ternary Cu^+2^ complexes. A series of complexes that have the property of efficiently mediating the cleavage of double-stranded DNA (plasmid pUC18 from calf thymus) through an oxidative mechanism have been synthesized [25,26,27,28]. Considering the promising results of the nuclease activity studies on isolated DNA fragments (plasmid pUC18), we have continued with the synthesis and characterization of new Cu^+2^ complexes and the evaluation of their antiproliferative, antioxidant and antibacterial potential, by performing in vitro and in vivo studies.

As a continuation of our research, we herein report the synthesis and characterization of two new Cu^+2^ complexes of N-sulfonamide ligands (HL1 and HL2, the structural formulae showed in Section 3). The complexes were structurally characterized and their antibacterial and cytotoxic activities were demonstrated.

## 2. Results and Discussion

### 2.1. Crystal Structures Description

Details with regard to crystal structures and refinement of both copper complexes are given in Table 1.

#### 2.1.1. Crystal Structure of [Cu_4_(L1)_4_(OH)_4_(DMF)_2_(H_2_O)] (C1)

The crystal and molecular structure of complex C1 was elucidated via single crystal X-ray diffraction and it was shown that the complex crystallizes in the non-centrosymmetric orthorhombic Pn space group. The asymmetric unit of C1 is rather complex and consists of four ligands, four water molecules, two recrystallization molecules of dimethylformamide (DMF) and four Cu^+2^ ions (Figure 1a). The embedded water molecules play an important role in the formation of crystal lattice.

Within the asymmetric unit, one Cu^+2^ ion (Cu4) forms a CuN_2_O_3_ entity which consists of one water and one DMF molecule, as well one nitrogen atom of adjacent thiadiazole ring. The coordination environment of a Cu4 metal atom comprises square pyramid geometry with the metal, two water molecules and two N_thiadiazole_ atoms in the basal plane and the carbonyl oxygen of dimethyl sulfoxide (DMSO) molecule at the tip of the pyramid. Two Cu^+2^ ions (Cu2 and Cu3) are involved in the formation of two CuN_2_O_3_ entities as well coordinating two water molecules, the same DMF molecule and two thiadiazole nitrogen atoms. Cu2 coordination environment depicts a distorted/tilted square pyramid configuration with the metal, two waters and two N_thiadiazole_ located roughly in the same basal plane and the tip of pyramid constituted by the oxygen of DMSO molecule. In the coordination of Cu3, the oxygen of DMSO molecule is located at the tip of the pyramid while the water molecules, the metal atom and N_thiadiazole_ atoms occupy positions shifted upwards and downwards from a common basal plane. Coordination geometry of Cu4 shows a heavily distorted square pyramid with the basal plane constituted by two N_thiadiazole_ and two waters with the metal shifted upwards with respect to the basal plane, while the tip of pyramid is given by the oxygen of DMSO molecule. Cu4 forms a CuN_2_O_2_ coordination geometry which contains two water molecules and two ligand thiadiazole nitrogen atoms but is not generating any particular shape; it can be regarded as a heavily elongated tetrahedral pyramid (Figure 2). All four ligands are participating in cooper coordination via the nitrogen atoms of thiadiazole rings to the detriment of deprotonated nitrogen atom of sulfonamide group which is consistent with other Cu^+2^ complexes [28,29]. This can be attributed to the charge delocalization in ligands in the thiadiazole rings and the sulfonamido groups. The coordination distances within asymmetric unit are situated in the range of 1.884 up to 2.782 Å, and angles of 69.01° to 116.05° (Table 2).

The displacements from the mean plane formed by the basal atoms towards the axial ligand in the square pyramid coordination sites are 2.595 Å for Cu2, 2.448 Å for Cu3 and 2.477 Å for Cu4.

The distances of the C=N double bonds within the four thiadiazole rings in the ligands are between 1.279 and 1.344 Å, which are specific for double bonds, the standard distance for such bonds being 1.25 Å. The single C–N bonds between phenyl and thiadiazole rings are between 1.326 and 1.341 Å and are shorter than the standard C–N distances of 1.416 Å. Additionally, within the thiadiazole rings, the N–N single bonds are between 1.386 and 1.400 Å and are shorter than the typical N–N distance of 1.42 Å. The C–S distances also within the thiaziazole rings are between 1.725 and 1.761 Å and are shorter than the typical C–S distances which are 1.819 Å. It was observed that the distances of the double bonds become slightly longer and those of the single bonds become smaller compared to the standard ones. This is due to the delocalization of electron density, this effect also being reported in other similar compounds [30]. Atoms from the sulfonamide component, together with the connecting carbon of the phenyl rings, form distorted tetrahedra. Thus, the angles for the O1-O2-C6-N1-S1 tetrahedron are between 104.3° and 117.3°; for the O3-O4-C14-N4-S3 tetrahedron, between 104.5° and 117.8°; for the O7-O8-C34-N10-S7 tetrahedron, between 105.7°–118.1° and for the O5-O6-C24-N7-S5 tetrahedron, the angles are between 105.3° and 118.1°. We recall that the value for the ideal tetrahedron is 109.47°.

The Cu–N and Cu–O distances in the C1 complex are comparable in length with those reported from other mononuclear Cu^+2^ complexes [31].

For the C1 complex, the coordination atoms are found in the trans position. The sum of deviations from 90° for all bond angles divided by four is 6.1° for Cu1. The sum of deviations from 90° for atoms in the basal plane is 3.64° for Cu2, 5.85° for Cu3 and 3.94 for Cu4.

Another way to characterize coordinated complexes is through structural parameters, τ4 and τ5, for four and five coordinated atoms, respectively.

Yang et al. [32] introduced the parameter, τ4, to distinguish whether the geometry of the coordination center is square planar or tetrahedral. For the four-coordinate Cu1 atom, the τ4 index has a value of 0.279 and has a geometry that tends towards square planar.

The trigonality index (τ5) of the C1 complex was calculated according to Addison et al. [33] and has the value 0 for a perfectly tetragonal geometry and 1 for a perfect trigonal bipyramid. For C1 complex, the following values were obtained: 0.424 for Cu3, 0.167 for Cu4 and 0.044 for Cu2, which suggests that the geometry of the complex is close to the tetragonal one.

Intermolecular interactions, which play a role in lattice cohesion and stabilizing, are represented by C-H···O, C-H···N and C-H···S contacts (Table 3). Recently, it was investigated and it was shown that C-H···S-type interactions have a significant contribution in solid-state stability [34]. An overall packing view along the oa-axis is presented in Figure 1b with the complex forming a polymeric arrangement supported by Cu⋯O coordination.

#### 2.1.2. Crystal Structure of [Cu(L2)_2_(phen)(H_2_O)] (C2)

It was determined via X-ray diffraction that the C2 complex (Figure 3) crystallizes in the centrosymmetric P2_1_/n space group of monoclinic crystal system with the asymmetric unit comprised of two deprotonated ligands (L2)^−^, one phenantroline from the reaction medium, one water molecule and one central Cu^+2^ ion. The coordination environment is CuN_4_O-type, with distorted square pyramid geometry in the basal plane consisting of two N_thiadiazole_ nitrogens of the two ligands and both N_phenantroline_ nitrogens while the tip of the pyramid is represented by the water molecule (Figure 4). The ligands (L2)^−^ coordinate via thiadiazole nitrogen atoms (N6) and (N2) to the Cu ion with Cu–N_thiadiazole_ distances of 1987 Å and 1.979 Å, respectively. The phenantroline to Cu distances, Cu–N_phenantroline_, are 2.014 and 2.064 Å. The distance between the central metal and the water molecule is 2.391 Å. Once again, like in the C1 complex, the deprotonated nitrogen of the sulfonamide group is not involved in coordination. In both investigated complexes, the Cu coordination to thiadiazole ring distances is similar to other crystals with thiadiazoles [35,36].

Unlike the C1 complex, in the case of the C2 complex, only two ligands are found. The distances for the C=N double bonds in the two thiadiazole rings are between 1.291 and 1.330 Å and are specific to double bonds, but slightly higher than the standard distance of 1.25 Å. The distances corresponding to the single C–N bonds between the phenyl and thiadiazole rings are 1.313 and 1.323 Å, being slightly shorter than in the C1 complex and also shorter than the specific C–N bond distances of 1.416 Å. Additionally, the simple N–N bonds are slightly shorter than those in the C1 complex, being 1.365 and 1.374 Å, respectively, and at the same time, shorter than the standard N–N distances of 1.42 Å. The single C–S bonds in the thiadiazole rings are between 1.728 and 1.771 Å and are comparable to those found in the C1 complex but shorter than the typical 1.819 Å distances for C–S bonds. As with the C1 complex, due to the delocalization of the charge within the thiadiazole rings, the double bonds become slightly longer, while the single bonds become slightly shorter. As with the first complex, the sulphonamide atoms and the carbon of the naphtalene adopt distorted tetrahedral geometries. For the tetrahedron, O1-O2-C10-N3-S2, the angles are between 105.8 and 116.9, and for the O3-O4-C29-N7-S3 tetrahedron, the angles are between 104.2° and 116.9°, being slightly distorted compared to the standard value of 109.47° in the case of a regular tetrahedron.

The distance from the plane formed by the basal atoms towards the axial ligand in the square pyramid is 2.484 Å.

For the C2 complex, the coordination atoms are found in the *cis* position. The average deviation from 90° of atoms from basal plane is 4.73°.

The trigonality index, *τ5*, of the C2 complex is 0.196, which means that the geometry of the complex is close to the tetragonal one.

Beside the intermolecular C-H···O, C-H···N, C-H···S interactions that occur within this complex and are also present in the case of the C1 complex, in this case, there are also C-H···π interactions participating in the crystal packaging (Table 3).

### 2.2. Hirshfeld Surfaces and Fingerprint Plots Analysis

Hirshfeld surface analysis is a useful tool used in order to visualize and to perform a quantitative and qualitative analysis of intermolecular interactions within crystals.

The molecular Hirshfeld surfaces can be interpreted based on red/blue/white color mapping as follows: intermolecular contacts with distances shorter than the sum of van der Waals radii (strong interactions) are mapped in red, interactions illustrated in white display distances approximately equal to the sum of van der Waals radii (weak interactions), while blue indicates intermolecular interactions which are characterized by distances longer than the sum of van der Waals radii.

The fingerprint plots are the 2D representations related to Hirshfeld surfaces where considering all (d_i_, d_e_) pairs, d_i_ represents the distance between nucleus of an atom inside the surface to the Hirshfeld surface while d_e_ represents exterior distance from the surface to an outside atom. Orange/red areas on fingerprint diagrams indicate a high density of (d_i_, d_e_) pairs; green, a moderate number, while blue suggests a lower density [37].

Figure 5 and Figure 6 present the Hirshfeld surfaces of complexes and their related fingerprint diagrams. Based on their analysis, the following statements can be made:

(i) The breakdown of fingerprint plots in individual contributions (%) is given in Table 4. It was observed that the H···H, H···O/H···H, H···C/C···H, H···N/N···H and H···S/S···H contacts show dominant percentages, which suggests that the crystal packings are governed by hydrogen bonds and van der Waals interactions.

(ii) The protruding spikes in the fingerprint plot of the C1 complex (Figure 5c) compared to the lack of spikes in C2 (Figure 6c) indicate that the C1 complex is characterized by shorter donor···acceptor distances for intermolecular interactions.

(iii) The C1 complex shows protruding Cu···O spikes, which indicates the importance of coordination and the formation of a polymeric supramolecular arrangement along the oa-axis, which is supported by coordinative interactions.

(iv) The large differences in the color-mapping range, −0.68 (red) to 1.9 (blue) for C1 compared to −0.07 (red) to 2.58 (blue) in C2, suggest that the C1 complex is more tighty packed.

### 2.3. Evaluation of Antibacterial Activity

Microdilution method was used to determine the antibacterial activity of the C1 and C2 copper complexes, HL1, HL2 free ligands and CuSO_4_ salt on six microbial strains, four Gram-positive (methicillin-susceptible *Staphylococcus aureus* (MSSA), methicillin-resistant *Staphylococcus aureus* (MRSA), *Staphylococcus lentus* and *Enterococcus faecium)* and two Gram-negative (*Escherichia coli* and *Pseudomonas aeruginosa).*

The antibacterial potential of the complexes was determined by evaluating the minimum inhibitory concentration (MIC) and minimum bactericidal concentration (MBC) values. The MIC, MBC and MIC index values for the two synthesized complexes were determined on six bacterial strains and are shown in Table 5.

The C1 complex showed bacteriostatic activity on *S*taphylococcus* aureus* MRSA and *Escherichia coli* (MIC index ≤ 4), bactericidal activity on *Staphylococcus aureus* MSSA (MIC index > 4) and no antibacterial activity against *Enterococcus faecium, Psudomonas aeruginosa* and *Staphyolococcus lentus.* Compared to the positive control, Gentamicin, the antibacterial activity of complex C1 is lower.

The C2 complex showed in vitro antibacterial activity against all selected bacterial reference strains. Thus, it showed bacteriostatic activity on *Staphyolococcus lentus, Escherichia coli* and *Pseudomonas aeruginosa* (MIC index ≤ 4) and bactericidal activity on *S*taphylococcus* aureus* MRSA, *Staphylococcus aureus* MSSA and *Enterococcus faecium* (MIC index > 4). The C2 presented antibacterial activity similar to that of Gentamicin.

The CuSO_4_ salt and the HL1 ligand are inactive on *Enterococcus faecium, Pseudomonas aeruginosa* and *Staphyolococcus lentus.* The antibacterial activity of CuSO_4_ on *S*taphylococcus* aureus* MRSA/MSSA and *Escherichia coli* is much weaker than that of complexes C1, C2 and HL2, and approximately equal to that of the HL1 free ligand. As we expected, the antibacterial activity of the C1 complex is better than that of the HL1 ligand. The findings were the same when comparing the activity of the C2 complex with the HL2 ligand.

A relevant finding of this study is the bactericidal activity of the C2 complex displayed against the *S*taphylococcus* aureus* MRSA reference strain. The *S*taphylococcus* aureus* MRSA strain is recognized worldwide as a prototype of multidrug resistance, and alternatives to classical antimicrobial agents are needed [38].

By coordinating the Cu^+2^ ion with sulfonamidic ligands, one can follow the modification of the biological activity of the ligands after complexation or can devise hypotheses concerning the reaction mechanism of the metallic ions or of the heterocyclic ligands. The active “component” of the complexes is not always easy to determine. In order to evaluate and to explain the biologic activity of a coordinative compound, one must take into account the contribution of the metallic ion, of the ligand, of the complex as an individual molecule or the synergic effects of two or three of these components. The preparation of some Cu^+2^ complexes with N-substituted heterocyclic sulfonamide ligands permitted the synthesis of some compounds with heightened antibacterial potential, covering a broader effective range [39]. As an explanation of the antibacterial activity of the tested complexes, we think that the character of the copper ions coordinated to the ligand may have its own role in the antimicrobial potential of the complexes, in addition to the chelation, which considerably increases the lipophilic character of the central metal ion because of the partial sharing of its positive charge with the donor groups and possible π-electron delocalization over the chelate ring. This favors the permeation through the lipid layer of the cell membrane. The increased liposolubility of the ligand upon being complexed with a metal may contribute to the easy transportation into the bacterial cells, which then blocks the metal-binding sites in the enzyme of microorganisms. Furthermore, different factors should be under consideration for metal complexes having antibacterial activities, *viz*, the chelate effect, the nature of the ligands, the total charge of the complex, and the nature of the ion that neutralizes the given ionic complex [21,40].

The antibacterial activity of Cu^+2^ complexes can be also explained by the fact that essential elements are denied to the microbial enzymatic systems of bacteria or that toxic metallic ions are provided to pathogenic agents. In both cases, the result is metallic chelates with high stability and inertia, which can disturb the metabolism of the microbial cell, leading to its final destruction [39].

Copper has different mechanisms of action that depend on the geometry of the complexes and the nature of the ligand [41,42]. Although the exact mechanism of the antimicrobial activity of copper is not known, many investigations have shown that ROS produced through Fenton-type reactions damages DNA. The release of copper ions causes the inactivation of enzymes that leads to its toxicity [43].

Studies have also revealed that Cu^+2^ complexes have been found to possess greater antimicrobial activity against both Gram-positive and Gram-negative bacteria compared to free sulfonamides [44]. Only the ionic form of free sulfonamides has an active antibacterial activity, but for its anionic form, the penetration efficiency across the lipoidal bacterial membrane is very low, which is due to its low lipophilicity. To enhance the permeation of the drug inside the cell, one possibility is to increase their lipophilicity via complexation of this kind of ligands with metal ions [44,45].

The most active sulfonamides contain a substitute of the amidic nitrogen atom of the sulfonamidic moiety. This substitute greatly influences the acid character and the bacteriostatic activity of the sulfonamide. An electron-withdraw substitute will induce a lower electron density around the amidic nitrogen atom (due to the-I effect), i.e., a stronger elimination of protons into the solution and an increase of the acid dissociation constant. At the same time, the electronegativity of the –SO_2_-group will increase, thus leading to maximal electron density of the group and to an optimal bacteriostatic activity of the sulfonamide [46,47].

Concerning our study, the antibacterial activity of the C1 and C2 complexes is relevant. The tested complexes showed antibacterial potential against both Gram-positive and Gram-negative bacteria. The C2 complex demonstrated superior antibacterial activity.

### 2.4. Cytotoxicity Assays

The cytotoxic potency of the C1 and C2 complexes was tested on three human cell lines, cervical carcinoma line (HeLa), colorectal carcinoma line (DLD-1) and a normal fibroblastic epithelial cell line (HFL1), while Cisplatin served as positive control. All cells were exposed at five different doses of complex ranging from 0.2 to 50 µM, at 24, 48 and 72 h. The same assay was carried out for HL1, HL2 free ligands and for CuSO_4_ salt.

The compounds showed overall significant inhibitory potency, the most active being the C2 complex. On the HeLa cell line, the IC_50_ of the C2 complex was 4-fold lower than the C1 complex, and that trend remains obvious for all time intervals. Compared to Cisplatin, the C2 complex was almost twice as active, while the C1 had a lower antiproliferative effect. Furthermore, the C2 complex was also active against the DLD-1 cells, IC_50_ of the C2 complex being at half of the C1, and also lower than Cisplatin, even though, in this case, the difference was less obvious. On the non-malignant fibroblastic cell line, HFL-1, the cytotoxic effect of the C2 complex was slightly lower than those of Cisplatin, but twice as high as the C1 complex. Notably, on the fibroblastic cell line, both compounds had a lower inhibitory effect compared to the malignant ones. Overall, those findings reveal a promising effect of the C2 complex; it was by far more potent that the C1 complex. Compared to Cisplatin, the C2 complex was more active on HeLa cells while in DLD-1, the difference was only marginal. Notably, both compounds were less toxic for the normal fibroblastic cell line, suggesting some selective toxicity against tumor cells (Figure 7).

The free ligands (HL1 and HL2) and the CuSO_4_ salt showed no significant cytotoxicity. So, the cytotoxicity is given by the complexes and the chelation of the ligands with the copper ion is essential for the cytotoxicity of the complexes.

The capacity of the new copper complexes to destroy DNA molecules depends on the type of ligand which is used in the coordination of Cu^+2^ ion. The ligands have two important roles in the biological activity of the complexes, namely, they influence the reactivity of the Cu^+2^ ion and also interact with DNA [48]. The sulfonamides used as ligands in the synthesized complexes increase the reactivity of Cu^+2^ ion and increase the biological activity of the complexes compared to Cu^+2^ salts. The presence of benzene, toluene, naphtalene in the sulfonamide structure is probably the main variable determining their capacity to destroy DNA. The complexes have a greater capacity to destroy DNA than the non-coordinated sulfonamide ligand. The plane aromatic rings allow the Cu^+2^ complex to come closer to DNA through intercalation between neighboring base pairs of DNA chains, and they link with them through π-stacking bonds [24]. A possible mechanism of action for the complexes is the oxidative degradation of the DNA molecule. The complexes interact with the DNA molecule through intercalation in the structure of the double helix. Immediately, the Cu^+2^ ion is reduced to Cu^+^ ion and in the presence of O_2_, the formed Cu^+^ complex produces reactive oxygen species in close proximity to the double helix. These species attack the 2-deoxyribose moiety, leading to the cleavage of the DNA chain. The biological activity of the copper complexes depends on the type of ligand used, a benzene, toluene or naphthalene ring. In the case of all the synthesized complexes, the ligand participates in the coordination through a single N_thiadiazole_ atom. The geometry of the complexes is similar. Thus, the differences among the biological activity of the complexes are due to the participation of the second type of ligand (in our case, the presence of phenantroline) in the coordination of the copper ion [29,49].

### 2.5. Evaluation of Apoptosis

Apoptosis assay by using the Annexin V-fluorescein isothiocyanate (FITC-A)/propidium iodide (PE-A) was carried out in order to investigate whether the growth inhibition is associated with apoptotic cell death. As presented in Figure 8, the lower-left quadrant represents the population of viable cells; they are negative to both FITC-A and PE-A; early necrotic cells are found in the lower-right quadrant, being positive for PE-A and FITC-A-negative. Apoptotic cells are found in the upper squares, which are FITC-A-positive; the upper-left quadrant (PE-A-negative) represents the early apoptotic cells, while the upper-right (PE-A-positive) represents the late apoptotic cells.

The figure represents scatter plots of Annexin V-fluorescein isothiocyanate (FITC-A) (*y*-axis) versus vital dye propidium iodide (PerCP) labeling (*x*-axes). Lower-left quadrants (absence of both markers) indicate viable cells; upper-left quadrants (FITC-A-positive, PerCP-negative) indicate the apoptotic cells (early-stage apoptosis), while the necrotic cells are scattered on the right side of the panel (PerCP staining alone or together with FITC-A).

In HeLa cells, the C1 complex induced a cellular death, mainly via necrosis, suggested by the pattern of the dot plot; at the highest concentration of 50 µM, the cells were distributed in both the upper-right quadrant (16.9%) and the lower-left quadrant (39.8%). In contrast, the C2 complex seems to be more specific in the induction of apoptosis, by increasing the percentage of late apoptotic cells to 87.6%, while the necrotic cells remain negligible. In DLD-1 cells, the apoptosis was less visible; at 50 µM, the C1 complex increased the early necrotic cells to 52.8%, while the C2 complex increased them to only 16.5%. Apoptosis seems to be the main mechanism of cell death in HFL1 cells, as the necrotic cells (lower-right quadrant) remained at a negligible percentage. At 50 µM, the C1 complex increased the late apoptotic cells (upper-right quadrant) from 5.6% in the control to 77.8%, while the C2 complex increased them to 87.4%.

In conclusion, we synthesized a copper complex, C2, which presents better cytotoxicity on HeLa and DLD-1 cell lines than Cisplatin, with less toxicity on normal fibroblastic cells than Cisplatin. The results are consistent with our apoptosis assay study.

## 3. Materials and Methods

All reagents were purchased from the Fluka (Europe) and Merck (Europe) chemical companies and were used without further purification.

Elemental analyses (C, N, H, S) were performed with a Vario EL analyser (Shimadzu Europa GmbH, Duisburg, Germany). IR spectra were recorded with a Jasco FT-IR-4100 spectrophotometer (JASCO, Heckmondwike, UK) using diffuse reflectance of incident radiation, focused on a sample, in the 4000–450 cm^–1^ range. Diffuse reflectance spectra and UV-Vis spectra of the complexes were recorded on a Jasco V-550 spectrophotometer (JASCO Labor-und Datentechnik GmbH, Pfungstadt, Germany) [50].

We synthesized two heterocyclic sulfonamide ligands, thiadiazole derivatives, using the Vogel method. The synthesis, structure and properties of the ligands, HL1 = N-(5-ethyl-[1,3,4]–thiadiazole–2-yl)-benzenesulfonamide) and HL2 = N-(5-(4-methylphenyl)-[1,3,4]–thiadiazole–2-yl)-naphtalenesulfonamide), have been reported in the articles published in the Polyhedron and Farmacia Journals [27,51]. The structural formulae of the synthesized sulfonamides used as ligands are presented in Figure 1.

### 3.1. Synthesis of the Complex, [Cu_4_(L1)_4_(OH)_4_(DMF)_2_(H_2_O)] (C1)

A total of 1 mmol of HL1 ligand was dissolved in a mixture of 50 mL methanol and 10 mL DMF. Separately, a solution of 0.5 mmol Cu(CH_3_COO)_2_·H_2_O and 5 mL of water was prepared. The copper solution was added to the ligand solution under continuous stirring. The resulting mixture was stirred at room temperature for two hours. The resulting light green solution was kept in a crystallizer at room temperature. After a few days, the complex crystallized as blue crystals, which can be used for crystallographic analysis by X-ray diffraction.

Anal. Calcd for (C1) C_46_H_54_Cu_4_N_14_O_14_S_8_ (MW = 1537.67 g∙mol^–1^): C, 35.90; H, 3.51; N, 12.75; S, 16.65%. Found C, 35.74; H, 3.98; N, 12.34; S, 16.38%. IR (KBr) ν_max_/cm^−1^: 1536 (ν(thiadiazole)); 1302 (ν_asym_ (S = O)), 1140 (ν_sym_ (S = O)), 932 (ν(S–N)). UV/Vis (solid) λ_max_/nm: 372 (π→π*), 418(LMCT), 622 (d-d). UV-Vis (DMF) (ε = 100 cm^−1^∙M^−1^) (Yield ca. 65%) (Appendix A: IR spectrum of the C1 complex and Appendix A: Diffuse UV-Vis reflection spectrum of the C1 complex).

### 3.2. Synthesis of the Complex [Cu(L2)_2_(phen)(H_2_O)] (C2)

A total of 1 mmol of ligand (HL2) was dissolved in 40 mL of methanol which was previously mixed with 2 mL of a NaOH solution 1 M. Separately, another mixture was prepared by dissolving 0.5 mmol CuSO_4_·5H_2_O and 0.5 mmol of phenantroline (phen) in 25 mL of methanol (a turquoise precipitate was obtained in the solution). By adding the ligand solution in the Cu^+2^-phenantroline mixture, the color turns to green. After stirring for an hour at a temperature of 35 °C, the reaction mixture was filtered in order to remove the precipitate (a complex of copper with phenantroline). After a few days, green crystals, suitable for X-ray diffraction analysis, formed in the solution.

Anal. Calcd for (C2) C_50_H_39_CuN_8_O_5_S_4_ (MW = 1023.67 g∙mol^−1^): C, 58.61; H, 3.81; N, 10.94; S, 12.50%. Found C, 58.48; H, 3.65; N, 10.82; S, 12.23%. IR (KBr) ν_max_/cm^−1^: 1460 (ν(thiadiazoles)); 1284 (ν_asym_ (S = O)), 1123 (ν_sym_ (S = O)), 938 (ν(S–N)). UV/Vis (solid) λ_max_/nm: 304 (π→π*), 423 (LMCT), 662 (d-d). UV-Vis (cacodylate buffer pH = 6)(ε = 105.6 cm^−1^∙M^−1^) (Yield ca. 65%) (Appendix A: IR spectrum of the C2 complex and Appendix A: Diffuse UV-Vis reflection spectrum of the C2 complex).

### 3.3. Solubility and Stability for the Complexes, C1 and C2, and the Ligands (HL1 and HL2)

The synthesized ligands (HL1 and HL2) have light colors and are stable in the presence of light and atmospheric oxygen. The compounds were insoluble in water, completely soluble in methanol and ethanol and very soluble in acetone, DMF, DMSO and tetrahidrofurane (THF).

Complex C1 is light and atmospheric oxygen stable. The yield of the reaction was of 65%. It is insoluble in methanol, ethanol, acetone and chloroform and soluble in DMF and DMSO.

Complex C2 is insoluble in water, ethanol, methanol, partially soluble in DMF and soluble in DMSO and THF. The compound is light and atmospheric-oxygen-stable as well.

The spectral study of C1 and C2 complexes in DMSO was performed at room temperature at different time intervals. The resulting results confirmed the stability of the complexes in DMSO at a concentration of 10^−5^ M for 48 h.

Additionally, the two complexes are soluble and stable in cacodylate buffer: DMF (1:39) (pH = 6), a mixture of solvents compatible with the biological environment, also used in the case of other Cu^+2^ complexes with ligands of the type N-substituted sulfonamides in order to test the nuclease activity of the complexes [25,26,27,28].

The stability of metal complex increases with decrease in size of metal cations. So, in the case of compounds formed with ions of transition metals from the first series and the same type of ligand, Cu^+2^ complexes have the highest stability. This is also demonstrated in the case of Cu^+2^ complexes with N-substituted sulfonamides [52,53].

### 3.4. X-ray Single-Crystal Diffraction and Structure Refinement

Suitable single crystals of the complexes were attached on a fine nylon loop and mounted on the goniometer of a SuperNova diffractometer (Malvern Panalytical, Malvern, UK) with the X-ray tube operating at 50 kV and 0.8 mA and equipped with two micro-sources (Cu and Mo) and Eos CCD detector. Data collection and the correction for Lorentz, polarization and absorption effects were carried out in CrysAlis PRO [54]. Crystal structures of complexes were solved with SHELXT method [55], using Intrinsic Phasing and refined via SHELXL [56] using least squares minimization in Olex2 software [57].

Hydrogen atoms were located and treated as riding, considering the isotropic displacement parameter U_iso_(H) = 1.2 U_eq_(C) for ternary CH groups [C-H = 0.93 Å], secondary CH_2_ groups [C-H = 0.97 Å] and 1.5 U_eq_(C), considered for all methyl CH_3_ groups [C-H = 0.96 Å]. Oxygen-bound hydrogen atoms were geometrically located and refined as riding.

### 3.5. Hirshfeld Surface and Related Fingerprint Plot Analysis

Hirshfeld surfaces and the corresponding 2D fingerprint plots were generated based on CIF files via the d_norm_ function [58]. CrystalExplorer software [59] was used for computation, considering the C–H, O–H bond lengths normalized to neutron diffraction distances.

### 3.6. Evaluation of Antibacterial Activity

#### 3.6.1. Minimum Inhibitory Concentration

The minimum inhibitory concentrations were determined using the microdilution method, according to the Clinical and Laboratory Standards Institute (CLSI) procedure [60]. The complex samples were diluted with DMSO. Four different concentrations were tested (0.2, 2, 12.5, 25 µg/mL), and as negative control, DMSO was used. Four bacterial reference strains (*n* = 3), methicillin-susceptible *Staphylococcus aureus* ATCC 25923 (MSSA), methicillin-resistant *Staphylococcus aureus* ATCC 700699 (MRSA), *Escherichia coli* ATCC 25922, *Pseudomonas aeruginosa* ATCC 27853, and two clinical isolates, *Staphylococcus lentus* and *Enterococcus faecium*, were used. The assessment was performed by using broth microdilution (twofold dilution) method on 96-well plates in triplicate. Briefly, 100  µL of Muller–Hinton (MH) broth was added to each well of the 96-well plate, stock solutions of the complexes (50 µg/mL) were prepared and 20  µL of bacterial suspension (1.5 × 10^6^ CFU/mL) was added in each well. Gentamicin was also included as standard antibiotic. The plates were incubated at 37 °C for 18  h. In order to evaluate bacterial growth/inhibition, after 18 h of incubation, 20  µL of MTT solution (3-(4,5-dimethylthiazol-2-yl)-2,5-diphenyltetrazolium bromide, 1.25  mg/mL) was added to each well. The plate was incubated for 1 h at 37 °C; bacterial growth was indicated by the appearance of purple color, and growth inhibition was indicated by a clear/yellow coloration in the well. All tests were performed in triplicate. The same method was carried out for HL1, HL2 free ligands and for CuSO_4_ salt. The MIC was defined as the lowest concentration of substance that completely inhibited the visible bacterial growth in the microdilution wells, compared to control wells (MH broth).

#### 3.6.2. Minimum Bactericidal Concentration

The minimum bactericidal concentration value, which represents the lowest concentration at which bacterial growth was completely inhibited, was also assessed. In order to evaluate MBC values, 100 µL of bacterial suspension was collected from the well where no visible bacterial growth was observed. The suspensions were inoculated on MH agar plates and incubated for 18 h at 37 °C. The MIC index was also calculated, based on the MBC/MIC ratio. Thus, an MBC/MIC ration ≤ 4 (MIC index ≤ 4) was considered bacteriostatic, and an MBC/MIC ration > 4 (MIC index > 4) was regarded as bactericidal [61].

### 3.7. Cell Culture and Cytotoxicity Assays

#### 3.7.1. Cell Culture

The cytotoxicity studies were conducted on three human cell lines, human cervical carcinoma line (HeLa) (ATCC, Manassas, VA, United States.), human colorectal carcinoma line (DLD-1) (ATCC) and a normal fibroblastic epithelial cell line (HFL1) (Sigma). The cells were seeded in DMEM (Sigma-Aldrich, Merck Romania SRL, Darmstadt, Germany), added with 10% FCS (Hyclone, Merck Romania SRL, Darmstadt, Germany) and 1 mM glutamine (Sigma-Aldrich). Penicillin-Streptomycin-Amphotericin B 1% antibiotic antimycotic solution 100× (Sigma-Aldrich) was added for prevention of microbial and fungi contamination. The cells were maintained in standard conditions, at 37 °C, 5% CO_2_, 95% relative humidity [28,29].

#### 3.7.2. Cytotoxicity Assays

The cytotoxicity was performed on 96-well plates (10^5^ cells/well). Twenty-four hours later, the medium was removed, and replaced by medium containing the complexes (C1 and C2) and Cisplatin (Ebewe Pharma Ges.m.b.H. Nfg. KG, Oberösterreich, Austria) as positive control, in concentrations of 50 µM, 25 µM, 12.5 µM, 2 µM and, respectively, 0.2 µM. The complexes were solubilized in DMSO, assuring the final maximum concentration in the medium was less than 1%. The negative control cells were seeded in the medium containing the same amount of DMSO as the complexes. MTT assay was performed at 24, 48 and 72 h using a spectrophotometer (Bio-Rad, Hercules, CA, USA) at 550 nm. The measurements were performed in triplicate each. The same assay was carried out for HL1, HL2 free ligands and for CuSO_4_ salt [28,29].

### 3.8. Evaluation of Apoptosis

The same cell lines were seeded on Petri glass plates (5 × 10^5^ cells/plate) in the same medium for 24 h. Then, the medium was removed and replaced by medium containing the C1 and C2 complexes in the same concentrations as for the cytotoxicity assay for another 24 h. Apoptosis was detected by using the Annexin V-fluorescein isothiocyanate (FITC) apoptosis detection kit, (BD Biosciences, San Jose, CA, USA) according to the manufacturer instructions. The flow cytometric analysis was performed using the BD FACS Canto II flow cytometer (Becton Dickinson & Company, Franklin Lakes, NJ, USA). A total of 10,000 events were recorded per sample; the fluorescence was detected using a 530/30 nm and a 575/26 nm band-pass (BP) filter. The flow cytometric detection differentiates between viable cells, Annexin V (−)/PI (−), early apoptotic cells Annexin V (+)/PI (−) and late apoptotic cells Annexin V (+)/PI (+) [29].

### 3.9. Statistics

Data are expressed as mean ± SD; the IC50 represents the concentration required to inhibit the cell proliferation by half; it was calculated from dose–response curve generated via non-linear regression using GraphPad Prism version 5.0 for Windows (GraphPad Software, San Diego, CA, USA).

## 4. Conclusions

Our research group studies were carried out with the intention of obtaining new complex combinations of Cu^+2^ with N-substituted sulfonamide ligands, with antitumor and antibacterial activities, with possible medical applications. For this purpose, we synthesized two new Cu(II) complexes with N-sulfonamide ligands, [Cu_4_(L1)_4_(OH)_4_(DMF)_2_(H_2_O)] (C1) (HL1 = N-(5-ethyl-[1,3,4]–thiadiazole–2-yl)-benzenesulfonamide) and [Cu(L2)_2_(phen)(H_2_O)] (C2) (HL2 = N-(5-(4-methylphenyl)-[1,3,4]–thiadiazole–2-yl)-naphtalenesulfonamide). The X-ray crystal structures of the complexes were determined, and the compounds were characterized via FT-IR and UV-Vis spectroscopy. The antibacterial and cytotoxic activities of the complexes were demonstrated. Both complexes present in vitro biological activities but the C2 complex is more active. The complexes have superior biological activity to the non-coordinated Cu^+2^ ion. In our opinion, the differences between the biological activities of the complexes are due to the participation of the second type of ligand (phenantroline) in the coordination of the copper ion.

## Data Availability

The data presented in this study are available within the article.

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
