# Peer review of "New Copper Complexes with Antibacterial and Cytotoxic Activity"

_ijms, 2023, doi:10.3390/ijms241813819_

Round 1

Reviewer 1 Report

The authors present in this paper the results of their research on the construction of complexes with sulfamide ligands. In general, this direction is a rather interesting area of research with a high potential for practical use of the obtained results. However, this manuscript is not carefully designed, the sections describing such important procedures as selection of conditions for synthesis of complexes, characterization of ligands, and explanation of their choice are unfortunately absent in the article. The introduction to the article and conclusions do not reflect the current state of research in this area, nor the interesting results obtained by the authors.

In addition, there are a number of comments to the manuscript: 

1) The second paragraph in the introduction should be rewritten because the authors' statement about cancer invading the body is not clear. To date, the mutational nature of cancer is still generally accepted. 

2) According to WHO statistics, the most common causes of death are diseases of the cardiovascular system, not cancer. It is also required to support each statement with relevant references. 

3) Also the second paragraph in the introduction: The authors state that 606,520 deaths and 1.8 million new cases of cancer are PROGNOSED by 2020. These data and projections need to be updated for 2023-2024. It would also be logical to present such forecasts not only for the USA, but also for the whole world or for example for Europe (where the authors are from). 

4) Taking into account the experimental character of the article, the authors should not delve into the history of cisplatin discovery. It would be more appropriate to present data on the synthesis and study of the biological activity of complex compounds of metals, and copper in particular.

5) Data on the synthesis of copper complexes, assessment of their toxicity, safety, biodistribution should be added to the introduction. The authors should also explain the choice of ligands used in their work for the complexes.

6) In Section 2. There are no data describing the synthesis of ligands, their formulas, selection of optimal conditions for the synthesis of ligands and complexes, and reaction schemes. It is necessary to supplement the section with the missing data.

7) Table 2. There is a missing point.

8) Figure 6. It is necessary to unify the font on the figures, increase the font size.

9) All abbreviations should be decoded at their first mention. 

10) Taking into account the potential use of the obtained complexes in biomedicine, it is necessary to add information on the solubility of the obtained complexes and their stability to hydrolysis in particular, as well as on the influence of ions on the stability of the obtained complexes.

11) It is required to give the MIC and IBC data of any other copper compound (even salt) as well as initial organic ligands to compare the obtained values.

12) It is also necessary to expand the part describing the potential mechanisms of antimicrobial action of the obtained complexes with the indication of the role in this process of both copper ions and ligands.

13) The data presented in Table 6 do not reflect the content of the section devoted to their description. It would be more logical and illustrative to present these data in the form of graphs.

14) Data on the cytotoxicity of copper salts (e.g. copper acetate) and on the cytotoxicity of the organic ligands themselves are required.

15) In the section devoted to the discussion of the biological activity of copper complexes, it is necessary to support each statement with an appropriate reference.

16) The conclusion should be considerably expanded by describing the main results obtained in this work with correlation with the data obtained by the authors, in particular with other copper complexes, analyzing also the influence of the type of ligand.

The text contains style errors and needs to be checked by a native speaker.

Author Response

Dear Reviewer,

First, thank you very much for your review and advice on our work. According to the comments, we have revised the manuscript as best as we can. Point-by-point responses to the comments are listed below. All the amendments were highlighted in red in the revised manuscript. It is worth mentioning that the comments and suggestions provided are very constructed and appreciated. Thanks again for your advice. We hope that the revised version of the manuscript is now acceptable for publication in IJMS journal.

Reviewer 1

The authors present in this paper the results of their research on the construction of complexes with sulfamide ligands. In general, this direction is a rather interesting area of research with a high potential for practical use of the obtained results. However, this manuscript is not carefully designed, the sections describing such important procedures as selection of conditions for synthesis of complexes, characterization of ligands, and explanation of their choice are unfortunately absent in the article. The introduction to the article and conclusions do not reflect the current state of research in this area, nor the interesting results obtained by the authors.

In addition, there are a number of comments to the manuscript: 

  • The second paragraph in the introduction should be rewritten because the authors' statement about cancer invading the body is not clear. To date, the mutational nature of cancer is still generally accepted. 

Answer: The changes made are presented in the manuscript in the introduction part.

  • According to WHO statistics, the most common causes of death are diseases of the cardiovascular system, not cancer. It is also required to support each statement with relevant references. 

Answer:The changes made are presented in the manuscript in the introduction part.

  • Also the second paragraph in the introduction: The authors state that 606,520 deaths and 1.8 million new cases of cancer are PROGNOSED by 2020. These data and projections need to be updated for 2023-2024. It would also be logical to present such forecasts not only for the USA, but also for the whole world or for example for Europe (where the authors are from). 

Answer:The changes made are presented in the manuscript in the introduction part.

  • Taking into account the experimental character of the article, the authors should not delve into the history of cisplatin discovery. It would be more appropriate to present data on the synthesis and study of the biological activity of complex compounds of metals, and copper in particular.

Answer:The changes made are presented in the manuscript in the introduction part.

  • Data on the synthesis of copper complexes, assessment of their toxicity, safety, biodistribution should be added to the introduction. The authors should also explain the choice of ligands used in their work for the complexes.

Answer:The changes made are presented in the manuscript in the introduction part.

  • In Section 2. There are no data describing the synthesis of ligands, their formulas, selection of optimal conditions for the synthesis of ligands and complexes, and reaction schemes. It is necessary to supplement the section with the missing data.

Answer:The synthesis, structure and properties of the ligand HL1 = N-(5-ethyl-[1,3,4]– thiadiazole–2-yl)-benzenesulfonamide) and HL2= N-(5-(4-methylphenyl)-[1,3,4]–thiadiazole–2-yl)-naphtalenesulfonamide), have been reported in the articles published in the Polyhedron and Farmacia Journals [Polyhedron 2010, Farmacia 2012]. We completed in manuscript this information. The structures of the ligands have been introduced in the manuscript (scheme 1). The synthesis conditions for C1 and C2 complexes are clearly presented in the manuscript in the Materials and Methods section (3.1 respectively 3.2)

7) Table 2. There is a missing point.

Answer: The missing point was added.

8) Figure 6. It is necessary to unify the font on the figures, increase the font size.

Answer: The figures were redrawn considering increased font size.

9) All abbreviations should be decoded at their first mention. 

Answer: Done.

10) Taking into account the potential use of the obtained complexes in biomedicine, it is necessary to add information on the solubility of the obtained complexes and their stability to hydrolysis in particular, as well as on the influence of ions on the stability of the obtained complexes.

Answer: We introduced in the manuscript the section:  3.3 Solubility and stability for the complexes C1 and C2  and for the ligands (HL1 and HL2)

11) It is required to give the MIC and IBC data of any other copper compound (even salt) as well as initial organic ligands to compare the obtained values.

Answer: We introduced in the manuscript at the section 2.3

12) It is also necessary to expand the part describing the potential mechanisms of antimicrobial action of the obtained complexes with the indication of the role in this process of both copper ions and ligands.

Answer: We introduced in the manuscript at the section 2.3

13) The data presented in Table 6 do not reflect the content of the section devoted to their description. It would be more logical and illustrative to present these data in the form of graphs.

Answer: Table 6 was replaced with Figure 7. See please section 2.4.

14) Data on the cytotoxicity of copper salts (e.g. copper acetate) and on the cytotoxicity of the  organic ligands themselves are required.

Answer: In Figure 7 the requested data are included.

15) In the section devoted to the discussion of the biological activity of copper complexes, it is necessary to support each statement with an appropriate reference.  

Answer: Done.

16) The conclusion should be considerably expanded by describing the main results obtained in this work with correlation with the data obtained by the authors, in particular with other copper complexes, analyzing also the influence of the type of ligand.

Answer:The changes made are presented in the manuscript in the conclusions part.

Reviewer 2 Report

The manuscript reports the synthesis and basic structural characterization of two copper complexes with antimicrobial and antitumor activity. The motivation for the studies resulted from the quest for new metallo-drugs which could be applied for cancer treatment. Next, the compounds were tested against selected Gram-positive and Gram-negative bacteria as well as their cytotoxicity was probed on two tumor cell lines. 

The article reports new results; however, additional information is needed in the manuscript and the data presentation requires correction so to better convey the intended message. In the following, please find my remarks regarding the current submission:

Despite mentioning in the abstract, the article report only shortened FT-IR and UV-Vis resultsPlease add registered spectra in supplementary information file.

Why do you use two different X-ray sources to obtain XRD data?

How does the determined Cu-N and Cu-O bond distances in C1 relate to the typical bond lengths of Cu-N and Cu-O in Cu mononuclear complexes? 

Can you determine the distortion parameter (calculated as the sum of deviations from 90° for all cis bond angles) for the reported structures? 

How large is the displacement from the mean plane formed by the basal atoms towards the axial ligand in the square pyramid coordination sites Cu2 – Cu4 in C1 and in Cu coordination polyhedron in C2?

The analysis of Hirshfeld plots is not easy when neighboring molecules (or their fragments) are not shown. It would be useful if at least of fragments of the neighbors are depicted.

The anti-microbial and antitumor activity tests were done for C1 and C2 complexes. Did you try to use Cu2+(with “bio-neutral” ligands) and the used N-substituted sulfonamides (or phenantroline) for comparison to try to evaluate if the biological activity should be ascribed to the complexes or to their fragments?

Some of the references seems not fully justified, in particular to works [51]-[54]. There are more than 20% of self-citations, e.g. to the other works of the first author.

Language needs correction, e.g. some words are misspelled (“cristal”), in Table 2 the first column title starts with dot, decimal separator should be “.”, not “,”.

Author Response

Dear Reviewer,

First, thank you very much for your review and advice on our work. According to the comments, we have revised the manuscript as best as we can. Point-by-point responses to the comments are listed below. All the amendments were highlighted in red in the revised manuscript. It is worth mentioning that the comments and suggestions provided are very constructed and appreciated. Thanks again for your advice. We hope that the revised version of the manuscript is now acceptable for publication in IJMS journal.

Reviewer 2

The manuscript reports the synthesis and basic structural characterization of two copper complexes with antimicrobial and antitumor activity. The motivation for the studies resulted from the quest for new metallo-drugs which could be applied for cancer treatment. Next, the compounds were tested against selected Gram-positive and Gram-negative bacteria as well as their cytotoxicity was probed on two tumor cell lines. 

The article reports new results; however, additional information is needed in the manuscript and the data presentation requires correction so to better convey the intended message. In the following, please find my remarks regarding the current submission:

  1. Despite mentioning in the abstract, the article report only shortened FT-IR and UV-Vis results. Please add registered spectra in supplementary information file.

Answer: Registered spectra were added as Supplementary Materials file. See please Figures S1-S4.

  1. Why do you use two different X-ray sources to obtain XRD data?

Answer: We prefer to use the tube with Cu radiation, but unfortunately a technical problem appeared with Cu tube and for this reason it was necessary to switch to Mo radiation.

  1. How does the determined Cu-N and Cu-O bond distances in C1 relate to the typical bond lengths of Cu-N and Cu-O in Cu mononuclear complexes?

Answer: The Cu-N and Cu-O distances in C1 are comparable in lengths with those reported from other mononuclear Cu(II) complexes. (Keypour, H.; Shayesteh, M.; Salehzadeh, S.; Dhers, S.; Maleki, F; Unver, Huseyn; Dilek, N. Probing the effect of arm length and inter- and intramolecular interactions in the formation of Cu(II) complexes of Schiff base ligands derived from some unsymmetrical tripodal amines. 39, 2015, 7429.)

  1. Can you determine the distortion parameter (calculated as the sum of deviations from 90° for all cis bond angles) for the reported structures?

Answer 1: For the C1 complex, the coordination atoms are found in the trans position. The sum of deviations from 90° for all bond angles divided by four is 6.1° for Cu1. The sum of deviations from 90° for atoms in the basal plane is 3.64° for Cu2, 5.85° for Cu3 and 3.94 for Cu4.

Answer 2: For the C2 complex, the coordination atoms are found in the cis position. The average deviation from 90° of atoms from basal plane is 4.73°.

Answer 3: Another way to characterize coordinated complexes is through structural parameters τ4 and τ5 for four and five coordinated atoms, respectively.

Yang et al. introduced the parameter τ4 to distinguish whether the geometry of the coordination center is square planar or tetrahedral.

C1: For the four-coordinate Cu1 atom, the τ4 index has a value of 0.279 and has a geometry that tends towards square planar. The trigonality index (τ5) of the C1 complex was calculated according to Addison et al. [y] and which has the value 0 for a perfectly tetragonal geometry and 1 for a perfect trigonal bipyramid. For C1 complex, the following values were obtained: 0.424 for Cu3, 0.167 for Cu4 and 0.044 for Cu2, what it suggests that the geometry of the complex is close to the tetragonal one.

Answer 4: C2: The trigonality index τ5 of the complex is 0.196, which means that the geometry of the complex is close to the tetragonal one.

All these additions were added to the manuscript.

  1. How large is the displacement from the mean plane formed by the basal atoms towards the axial ligand in the square pyramid coordination sites Cu2 – Cu4 in C1 and in Cu coordination polyhedron in C2?

Answer:  In C2 the distance from plane formed by the basal atoms towards the axial ligand in the square pyramid is 2.484 Å.

Answer:  In C1 the displacements from the mean plane formed by the basal atoms towards the axial ligand in the square pyramid coordination sites is 2.595 Å for Cu2, 2.448 Å for Cu3 and 2.477 Å for Cu4.

  1. The analysis of Hirshfeld plots is not easy when neighboring molecules (or their fragments) are not shown. It would be useful if at least of fragments of the neighbors are depicted.

Answer: We chosed this simplified approach to avoid very crowded figures because there are many contacts with distances smaller than the sum of the vdW radii.

  1. The anti-microbial and antitumor activity tests were done for C1 and C2 complexes. Did you try to use Cu2+(with “bio-neutral” ligands) and the used N-substituted sulfonamides (or phenantroline) for comparison to try to evaluate if the biological activity should be ascribed to the complexes or to their fragments?

Answer: We used CuSO4 salt, HL1 and HL2 free ligands. The results are presented  in the manuscript at the section 2.3

  1. Some of the references seems not fully justified, in particular to works [51]-[54]. There are more than 20% of self-citations, e.g. to the other works of the first author.

Answer: The self-citations [51]-[54] were deleted. See please references.

Reviewer 3 Report

Decision:

Minor revision

Comments
          The authors reported
New Copper complexes with antibacterial and cytotoxic activity: However, the authors should address the following points outlined below to improve the scientific quality. After the suggested revisions are carefully addressed, this work may be considered for publication.

1.     Author should carefully check the manuscript for language errors. There are several grammatical mistakes.

2.     In the abstract section, author need to highlight only the important points and finding of this work.

3.     In introduction section author should focus why these complexes are better choice then other metals, and metals oxides.

 https://www.mdpi.com/2073-4352/12/1/72

https://www.nature.com/articles/s41598-023-28356-y

4.     Author should explain in the introduction how their materials is better for antibacterial properties then previous research.

5.     In conclusion author should few more important points of this research

1.     Author should carefully check the manuscript for language errors. There are several grammatical mistakes.

Author Response

Dear Reviewer,

First, thank you very much for your review and advice on our work. According to the comments, we have revised the manuscript as best as we can. Point-by-point responses to the comments are listed below. All the amendments were highlighted in red in the revised manuscript. It is worth mentioning that the comments and suggestions provided are very constructed and appreciated. Thanks again for your advice. We hope that the revised version of the manuscript is now acceptable for publication in IJMS journal.

Reviewer 3

Comments
          The authors reported New Copper complexes with antibacterial and cytotoxic activity: However, the authors should address the following points outlined below to improve the scientific quality. After the suggested revisions are carefully addressed, this work may be considered for publication.

  1. Author should carefully check the manuscript for language errors. There are several grammatical mistakes.

Answer: The article was read and corrected by a native English speaker.

  1. In the abstract section, author need to highlight only the important points and finding of this work.

Answer: In the abstract section are highlighted the important points and finding of the work.

  1. In introduction section author should focus why these complexes are better choice then other metals, and metals oxides.

https://www.mdpi.com/2073-4352/12/1/72

https://www.nature.com/articles/s41598-023-28356-y

Answer: The changes made are presented in the manuscript in the introduction part.

  1. Author should explain in the introduction how their materials is better for antibacterial properties then previous research.

Answer:The changes made are presented in the manuscript in the introduction part.

  1. In conclusion author should few more important points of this research.

Answer:The changes made are presented in the manuscript in the conclusions part.

Round 2

Reviewer 1 Report

1. References 6 and 7 are completely irrelevant to this paper and are about the synthesis of bimetallic or metal oxide nanoparticles, so they should be removed from the list of references cited. 

2. Unfortunately, the column data in the diagrams presented in Figure 7 are almost unreadable, because in black and white color the columns do not contrast well with each other. It is required to correct the design of this figure.

3. it is necessary to check the correctness of writing formulas in the text, in particular on page 12.

Author Response

Dear Reviewer,

Thank you very much for your review and advice on our work. According to the comments, we have revised the manuscript as best as we can. Point-by-point responses to the comments are listed below. All the amendments were highlighted in red in the revised manuscript. We hope that the revised version of the manuscript is now acceptable for publication in IJMS journal.

Reviewer 1

  1. References 6 and 7 are completely irrelevant to this paper and are about the synthesis of bimetallic or metal oxide nanoparticles, so they should be removed from the list of references cited. 

Answer: References 6 and 7 were replaced with relevant references.

  1. Unfortunately, the column data in the diagrams presented in Figure 7 are almost unreadable, because in black and white color the columns do not contrast well with each other. It is required to correct the design of this figure.

Answer: We corrected the design for Figure 7.

  1. It is necessary to check the correctness of writing formulas in the text, in particular on page 12.

Answer: We checked the correctness of writing formulas in the manuscript.

Reviewer 2 Report

The article was greatly improved during the revision process. In my opinion it is now publishable as is.

Author Response

Reviewer 2

Dear Reviewer,

We would like to thank you for taking the time to review our manuscript and provide us with positive and encouraging feedback. We appreciate your support.

The article was greatly improved during the revision process. In my opinion it is now publishable as is.
